# How Cover Crop Sowing Date Impacts upon Their Growth, Nutrient Assimilation and the Yield of the Subsequent Commercial Crop

Paul Cottney [1,2,*] , Lisa Black [1], Paul Williams [2] and Ethel White [1,†]

1   Agri-Food and Biosciences Institute, Crossnacreevy BT6 9SH, UK; Lisa.Black@afbini.gov.uk (L.B.); Ethel.whiteni@gmail.com (E.W.)
2   Institute for Global Food Security, School of Biological Sciences, Queen's University Belfast, Belfast BT9 5DL, UK; p.williams@qub.ac.uk
*   Correspondence: pcottney02@qub.ac.uk; Tel.: +44-028-9054-8007
†   The author has retired.

**Abstract:** Cover crops are typically sown post-harvest of commercial crops, prior to winter, which means that as sowing date is delayed, so will biomass production potential. The wide range of benefits associated with cover crops relies on them to produce sufficient biomass. Therefore, it must be identified how late certain species of cover crops can be sown. In the climatic conditions of Northern Ireland, not only has no research been conducted on how cover crops perform at various sowing dates but also their effect on the subsequent commercial crop yield has not been investigated. Addressing these issue will in turn help provide recommendations to maximise and encourage later sowing of cover crops. Consequently, five species of cover crops were chosen, from a range of families, then sown on 14 August, 7 September and 27 September. This is to mimic when land becomes fallow post-harvest of typical crops/rotations to this region. It was found that tillage radish (*Raphanus sativus* L.), when sown on the earliest date, could accumulate a maximum of 261 kg/ha of nitrogen (N), whereas, when sown on the last date, phacelia (*Phacelia tanacetifolia* L.) significantly outperformed all other species and assimilated 70 kg/ha of N. The cover crops were then incorporated into the soil and over-sown with spring barley (*Hordeum vulgare* L.). However, the spring barley yield was unaffected by any treatments. This trial shows that the non-leguminous species chosen are highly effective in assimilating nutrients when sown mid-August until early-September.

**Keywords:** nitrogen assimilation; spring barley yield; weed management; carbon assimilation; biofertiliser; light interception

## 1. Introduction

When the sowing of cover crops is delayed later into autumn, their exposure to conditions critical for growth is reduced. This is detrimental to cover crop growth rate due to a decrease in the average ambient air and soil temperature as well as a continuously diminishing quality and quantity of light. In addition, soil moisture will typically increase and can negatively impact on soil trafficability which is critical for machinery to operate and sow the cover crop. Sowing early post-harvest of the previous commercial crop is important for biomass growth and also for nutrient accumulation [1] and grazing potential [2,3]. Early sowing of cover crops is the optimum practice to maximise their benefits but this is not always possible as their sowing date depends on the harvest date of the prior crop [1]. Early harvested crops (mid-July to early-August), e.g., winter barley (*Hordeum vulgare* L.), are more conducive to sowing early but these crops do not account for all rotations that have fallow land over winter prior to a spring crop. Therefore, to maximise the area of land sown with cover crops, they must also be planted after crops with harvests that are later than winter barley, such as winter wheat (*Triticum aestivum* L.), spring barley (*Hordeum*

*vulgare* L.) and vegetables. This raises the need for investigation of how late in the autumn, in the climatic conditions of Northern Ireland (NI), can cover crops be sown and grow successfully, and which species are better suited to later sowing dates.

Different species of cover crops exhibit variances in how they perform under delayed sowing, due to their vigour and competitiveness [4]. Producing high biomass increases nutrient assimilation [5]. Therefore, delayed sowing will reduce effectiveness in mitigating against N leaching [6]. This is because the potential to scavenge N has been found to relate to leaf expansion rate along with radiation interception [6] and the depth and density of the roots [5]. Delayed sowing of cover crops can also affect weed suppression ability [7]. This is important as an integrated management tool for growers, especially those that are organic, where identification of the best species and management practices to suppress weeds is essential [8]. Brust et al. [9] found that forage radish (*Raphanus sativus var. oleiformis* L.) was consistent in reducing weed biomass at all test sites (3 across 2 years in Germany) and that phacelia reduced weed dry matter by 77% relative to fallow. These authors noted that fast competitive growth boosted light interception, enhanced the plant canopy and, therefore, increased ability to shade out weeds.

Cover crops add carbon (C) to soils, but this benefit will also be reduced with delayed sowing, due to the decline in biomass which may also reduce the C concentration within the plant, as more mature early-sown crops may have more C-rich structural compounds. Delayed sowing, therefore, could affect the carbon:nitrogen (C:N) ratio [8,10], which influences the rate of N mineralisation [11,12]. The speculated effect would be a decrease in C:N due to a lower proportion of structural C compounds within the plants. This decrease in C:N ratio could result in a higher mineralisation of nutrients from the cover crop biomass which could better supply the commercial crop with N. However, less N accumulated by the cover crop could result in more N leached over winter, which could reduce the quantity of N available to the commercial crop, in comparison to earlier sowing. This in turn could affect spring barley grain yields, if N fertiliser rate is insufficient.

*Objective*

From a list of sixteen species of cover crops investigated by Cottney et al. [13], five species have been chosen from a range of families which showed potential to increase nutrient cycling and grain yield in that greenhouse experiment. The chosen species include forage rape (*Brassica napus* L.), tillage radish (*Raphanus sativus* L.), vetch (*Vicia villosa* L.), westerwolds (*Lolium multiflorum* L.) and phacelia (*Phacelia tanacetifolia* L.) to be investigated under field conditions. The species will be sown at three different dates to represent sowing of cover crops after harvest of winter barley, winter wheat/spring barley, and to represent a delayed commercial crop harvest. The objective is to investigate the effect on:

(1)    Cover crop biomass,
(2)    Cover crop nutrient assimilation, and
(3)    Consequence on spring barley yield supplemented with a reduced rate N program (70 kg/ha of inorganic N).

## 2. Materials and Methods

### 2.1. Experimental Design

A split-plot experimental design was generated using the procedure AGHIERARCHICAL [14] with two treatments replicated across 8 blocks during cover crop growth. The experiment was reduced in replication to 4 blocks during spring barley growth due to resource limitations. Treatment factors included planting date (whole plot) and species (sub-plot), each of which had 3 and 6 levels, respectively. The whole plot was completely randomised, with the sub-plots randomised within the whole plots. An unplanted control of bare fallow and the 5 species of cover crops (vetch, forage rape, tillage radish, westerwolds and phacelia) were sown, as shown in Appendix (Table A1). At each sowing date, the control was cultivated with the disc and left unsown as a bare fallow. This mirrors farmer practice of stale seedbed creation whereby fallow land may be cultivated to both

destroy and encourage more weeds to grow. Hence, each control at each sowing date is different to each other where each control will be presented separately in the data.

The experiment was planned to be repeated for two years but the second year replication was not possible due excessive rainfall.

### 2.2. Location

The trial site was located in Hillsborough, Co. Down, NI (latitude 54.445117, and longitude 6.096430). The soil is a clay loam to 30 cm with a particle distribution analysis of 44.7% sand, 33.5% silt and 22.9% clay in a 15 cm profile, with a deep clay after 35 cm in the profile. The soil analysis is shown in Table 1, with soil temperatures, ambient air and average site rainfall shown in Figures A1–A3, respectively. Soil and ambient air temperatures were logged using a Tinytag Plus 2 TGP-4510 datalogger and a soil probe (PB-5001) (West Sussex, UK) measuring to a soil depth of 15 cm.

**Table 1.** Standard soil test and soil mineral nitrogen (SMN) prior to sowing the cover crops.

| Parameter | Unit | Value |
|---|---|---|
| pH | | 5.87 |
| Phosphorous (P) | mg/L | 58.2 |
| Potassium (K) | mg/L | 202.5 |
| Magnesium (Mg) | mg/L | 180.3 |
| Sulphur (S) | mg/L | 18.3 |
| Total soil N | % | 0.34 |
| Total soil C | % | 3.68 |
| Nitrite ($NO^-_2$) + nitrate ($NO^-_3$) | mg/kg | 14.9 |
| Ammonium ($NH^+_4$) | mg/kg | 14.9 |
| Total SMN * | mg/kg | 29.9 |

* Bulk density of 1.2 g/cm$^3$.

### 2.3. Crop Management

The previous crop was whole-cropped spring barley, with the prior rotation having been all cereals for 8 previous years. Additional nutrients were supplied in the form of slurry to remove this as a limiting factor to cover crop growth, using 35 m$^3$/ha of pig slurry (finisher) (Table A2). The slurry was applied on 3 August by tanker with a dribble bar attached and metered using a flow sensor. A Lemkin disc Heliodor disc (Alpen, Germany) was used to cultivate prior to each sowing date to a depth of 10 cm. Cover crops were then sown using a Wintersteiger plot sower (Essex, England) following the recommended sowing rates advised by RAGT Seeds (Wilson, 2018, personal communication) (A 4.1). Sowing dates (SD) were 14 August 2018 (SD 1), 7 September 2018 (SD 2), and 27 September 2018 (SD 3).

A Bomford flail (Worcestershire, UK) was used to mechanically mulch, and thus terminate, the cover crops on 28 February 2019. The plots were then mouldboard ploughed (Kverneland, Merseyside, UK) to 20 cm on 11 April 2019. A seedbed was created by using a power-harrow (parallel to the plots) (Kverneland, Merseyside, UK) and the spring barley variety KWS Irina was sown to establish 325 seeds/m$^2$ accounting for the TGW (thousand grain weight), its% germination and field losses, using a Wintersteiger plot sower on 19 April 2019. Plot size during cover crop growth was 1.68 m × 16 m and 1.68 m × 12 m yielded from the subsequent plots of spring barley. The spring barley received a spray programme to control weeds, pests and diseases, with 70 kg/ha of inorganic N applied (Table A3) using a Sissis high accuracy fertiliser applicator on 23 May 2019. Plots were yielded using a Sampo plot harvester combine (Pori, Finland) on 18 September 2019. Prior to harvest, plots were visually assessed and scored for percentage crop damage of lodging, leaning, necking and brackling, as well as chickweed growth through the spring barley, which was scored on a 1–9 scale (1 = the highest, 9 = the lowest).

### 2.4. Data Collection

#### 2.4.1. Soil Mineral Nitrogen (SMN)

Initial soil samples from each block were sampled to a depth of 15 cm on 2 August 2018. On 25 February 2019, the control plots with nothing planted were sampled to 15 cm depth for soil mineral nitrogen (SMN). Fifty grams of soil sieved to 4 mm was mixed with 100 mL of 2 M KCl, using an additional, 10 g of the soil sample to obtain the dry matter. The mix of soil and KCl was shaken in an orbital shaker for 1 h at 200 RPM, centrifuged at 2970 g for 4 min and the liquid fraction filtered through No. 40 Whatman filter paper. Two blanks were run with each set of extractions to determine and adjust for any contamination. Soil N was transformed into kg/ha using a bulk density of 1.2 g/cm$^3$ and multiplying by the sampled depth of 15 cm.

#### 2.4.2. Leaf Area Index (LAI)

Ceptometer readings were taken using an AccuPAR LP-80 (METER Group, Inc. Pullman, Washington, DC, USA) which calculated the leaf area index (LAI) using a model (documented in the manual). Measurements were taken monthly, on the same sub-plots when weather conditions allowed (readings required dry and bright conditions resulting in a variation in measurement date). The LAI is an effective measurement which is quick and non-destructive to demonstrate crop growth, senescence and damage by frost over time. The ceptometer measures light levels above and below crop and builds a LAI which is the amount of green cover (metres squared) per metre of ground area. Differences in the model's accuracy will exist between species as each cannot be assumed as homogenous in building this estimation of plant cover. However, it does provide a low-cost method of comparing crop growth over time using non-invasive techniques.

#### 2.4.3. Biomass Sampling

Cover crop biomass was determined on 4 February 2019 using a 0.71 × 0.71 m quadrat. The brassicas (tillage radish and forage rape) had the roots extracted and washed under a tap as this biomass was deemed to be a large proportion of the total biomass. The weeds were separated from cover crop biomass. The roots were washed under a tap to remove soil. All biomass fractions were weighed, chopped to 4 cm using a stainless steel knife and 100 g subsamples taken. The subsamples were washed with deionised water, then dried at 60 °C for 48 h until a constant weight.

#### 2.4.4. N and C Determination

Dried samples of the cover crop above-ground biomass, roots and weeds were milled to 1.0 mm using a Cyclotec 293 mill (FOSS, Cheshire, UK). N and C were analysed using the Dumas dry combustion method with a Trumac CN analyser (Leco Corporation, Michigan, USA) furnace temperature 1350 °C, with quality controls of an in-house verified reference material run every 20 samples. Nutrient accumulation was calculated through multiplication of the relative nutrient% by biomass. Nutrient accumulation of the weeds was generated by multiplying the weed biomass of each plot by the average nutrient concentration of the weed biomass from the relative controls of SD 1, SD 2 and SD 3.

#### 2.4.5. Energy Dispersive X-Ray Fluorescence (EDXRF)

EDXRF was used to measure a broad-spectrum nutrient profile in a select list of the best performing cover crops. Samples were milled to 1 mm (as described in Section 2.4.4) with 2.5–3.5 g (depending on species) loaded into sample cups to a depth >4 mm. To create a pellet, 300 PSI was applied for 20 s. A certified reference material (mixed Polish herbs INCT-MPH-2) was used in each batch of sample, allowing recoveries to be detected and coefficient of variation (CV) to be gauged. Only recoveries of 100 +/− 20% with a maximum CV of 10% were used as parameters to accept the specific nutrients from the profile measured. Nutrient uptakes were calculated through multiplication of cover crop

biomass by its relative nutrient concentration. The nutrient accumulation of the weeds was not added on to the results shown.

### 2.5. Statistical Analysis

Genstat Version 18 [14] was used to analyse parameters of cover crop growth. Restricted maximum likelihood (REML) was used to analyse the cover crop nutrient accumulation and spring barley yield due to the unequal number of observations as REML produces predicted means.

The ceptometer measurements were analysed using REML as the monthly data were correlated. To account for unequally spaced measurements, the power model of order 1 was applied to the random component (Day Number). The fixed model included was Day Number + Sowing date + Species + Day Number × Sowing date + Day Number × Species + Sowing date × Species + Day Number × Sowing date × Species, and random components were Rep + Rep × Whole plot + Rep × Whole plot × Sub-plots + Plot × Day Number.

Fisher's unprotected post hoc analysis was applied to discriminate differences between species. Grain yield had covariates included in the REML analysis to adjust yield due to crop damage from chickweed, lodging and brackling which were scored on a plot basis prior to harvest. REML analysis of repeated measures was used to analyse the ceptometer readings of LAI with power model of order 1 applied to the random component of Day Number to take account of the correlation structure of the unequally spaced repeat measurements.

Results are deemed significant if probability due to random chance is under 5% ($p < 0.05$) and tendencies are regarded under 10% ($p < 0.10$).

## 3. Results

### 3.1. Leaf Area Index (LAI)

LAI was significantly affected by both sowing date ($p < 0.001$) and species ($p < 0.001$) and exhibited significant interaction ($p < 0.001$) (Table 2). The LAI indirectly shows the natural senescence and the degree the species were affected by frost. Species most affected by the frost/winter temperatures experienced include tillage radish and the phacelia as seen by the large LAI declines in SD 1 and SD 2 between November and December (Figures 1 and A2). Westerwolds exhibited the largest LAI, but westerwolds sown at SD 2 exhibited the second lowest LAI.

LAI on 16 October was higher in all species sown at SD 1, than from the two later sowings. By mid-November, all species, except westerwolds, sown on SD 2 had increased LAI/interception. By mid-December, LAI had decreased in all species sown on SD 1 and were similar to those of the species sown on SD 2. Changes in LAI between mid-December and mid-February of the first two sowing dates were relatively small, decreasing for most species. All species, when sown later (SD 3, 27 September 2018) had low LAIs.

**Table 2.** REML analysis of leaf area index (LAI).

| Treatment | N.D.F * | Chi Probability | LSD |
|---|---|---|---|
| Day Number | 3 | <0.001 | 0.248 |
| Sowing Date | 1 | <0.001 | 0.41 |
| Species | 5 | <0.001 | 0.308 |
| Day Number × Sowing Date | 3 | <0.001 | 0.494 |
| Day Number × Species | 15 | <0.001 | 0.615 |
| Sowing Date × Species | 5 | <0.001 | 0.576 |
| Day Number × Sowing Date × Species | 15 | <0.001 | 1.061 |

* Number of degrees freedom.

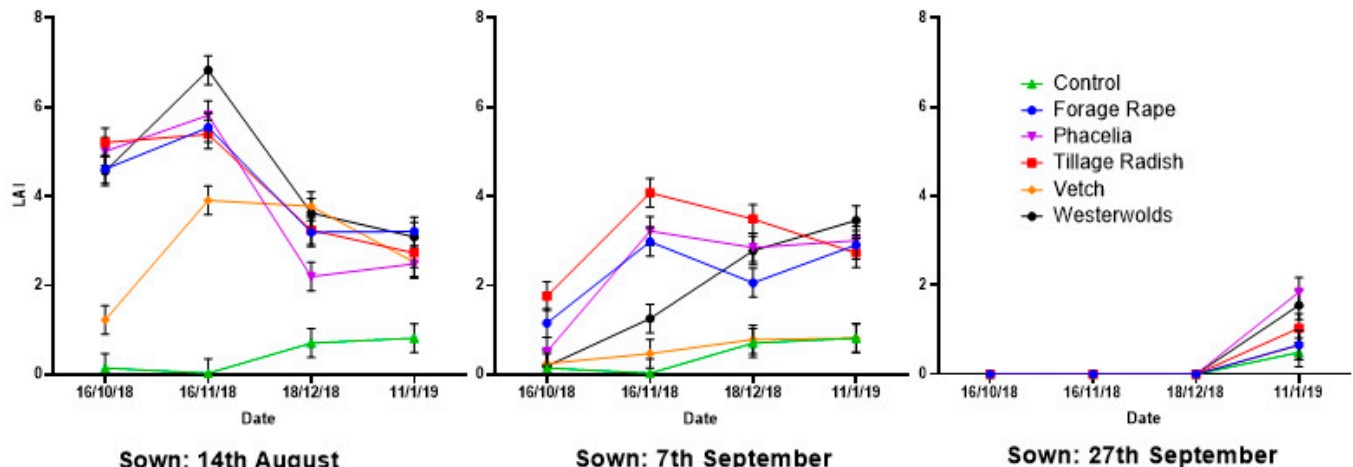

**Figure 1.** Leaf area index (LAI) of the three different sowing dates for each cover crop measured at the monthly dates. Error bars represent the standard error of the mean (SEM). Early-sown crops were more affected by frosts than later sowing. N = 8 for each mean.

### 3.2. Biomass Production

Figure 2 shows the fractions of biomass produced by the cover crops at the different sowing dates. The letters show Fisher's unprotected post hoc LSD (0.05) for total production inclusive of cover crop, roots and weeds. Total biomass was affected by sowing date ($p < 0.001$) and species ($p < 0.001$), and exhibited a significant interaction ($p < 0.001$) (Table A4). Biomass production in all species decreased with later sowing. The extent of the decrease varied with species, hence the significant interaction. Tillage radish and forage rape produced the greatest overall biomass at SD 1 ($p < 0.05$) (Figure 2), with tillage radish producing 6447 kg/ha DM including roots and the forage rape producing 6026 kg/ha including roots.

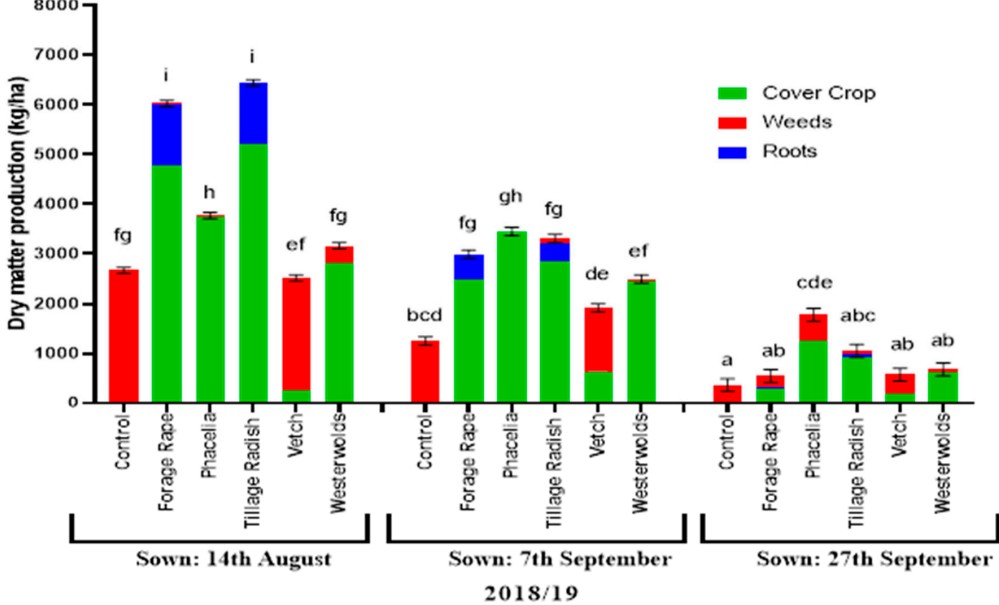

**Figure 2.** A stacked bar chart to show the biomass produced from above ground cover crops, their roots and also the weeds which grew in each treatment. Error bars represent the standard error of the mean (SEM). Means followed by a common letter are not significantly different by Fisher's unprotected post hoc least significant difference (LSD) at the 0.05% level of significance which equals 884.7. N = 8 for each mean.

Between SD 1 and SD 2, phacelia did not display a significant reduction in biomass—3732 and 3464 kg/ha (excluding weeds), respectively. At SD 2 and SD 3, of all the species, phacelia produced the greatest levels of biomass. At SD 3, phacelia produced 1830 kg/ha, almost two-fold more biomass than tillage radish (967 kg/ha). The vetch was the lowest biomass-producing cover crop at any date. Delaying planting from SD 1 to SD 2, numerically increased the vetch biomass from 302 to 728 kg/ha. Vetch and westerwolds produced similar levels of biomass at both SD 1 and SD 2, but the vetch produced significantly less cover crop biomass and significantly more weeds. It must also be noted that late-sown forage rape was the only species that exhibited crop damage caused by pigeons, which reduced biomass.

### 3.3. Roots

As would be expected, root biomass was substantial in tillage radish and forage rape, hence its measurement in these species. Total roots biomass in these species was affected by sowing date. SD 1 led to the largest accumulation of over 1200 kg/ha of each species, which declined to 429 kg/ha for tillage radish and 587 kg/ha for forage rape at SD 2 to less than 100 kg/ha of root biomass recorded at SD 3, for either species.

### 3.4. Weeds

Overall weed growth (measured from control plots) was 2744, 2271 and 717 kg/ha at SD 1, SD 2 and SD 3, respectively. Weed growth was affected by sowing date ($p < 0.05$), where later sowing reduced weed biomass found in the cover crops (Table A4). This declined on average from 534 to 285 kg/ha and finally to 211 kg/ha at SD 1, SD 2 and SD 3, respectively. Vetch was the only species that did not suppress weeds ($p < 0.05$). A significant interaction between sowing date and species was found ($p < 0.001$) (Table A4).

### 3.5. Cover Crop% N

N concentration (%) was not affected by sowing date, but species exhibited a significant difference ($p < 0.001$) and a significant interaction ($p < 0.01$) (Table A4). Although sowing date was not significant, a significant interaction ($p < 0.01$) with species was exhibited (Table A4.) This suggests that there must have been cross over of means, caused by treatments, as outlined by Grace-Martin [15]. The control exhibited a significantly lower ($p < 0.05$) N concentration (2.6%) in comparison to all other species (means not presented). The vetch had the highest% N of 5.15% when planted on SD 1 but when planted at the latest sowing date it did not have any significantly higher% N than any of the controls of just weeds.

### 3.6. N Accumulation (Biomass x% N)

SD 1 resulted in the largest N accumulation ($p < 0.001$) for all species, including the control (Table A4). A maximum accumulation of 261 kg N/ha occurred in tillage radish, with forage rape accumulating a similar amount of 255 kg N/ha. At SD 2, tillage radish accumulated 162 kg N/ha, which was the largest at that date compared to the control which accumulated the least N (25 kg/ha). At SD 3, phacelia outperformed all other cover crops and accumulated 70 kg N/ha, whilst N uptake in tillage radish and forage rape decreased further than the other species. When weed N accumulation was subtracted from the vetch N accumulation, this species only accumulated 18 kg N/ha at SD 1, 25 kg N/ha at SD 2 and 6 kg N/ha at SD 3, further reflecting the low biomass production seen in Figure 3.

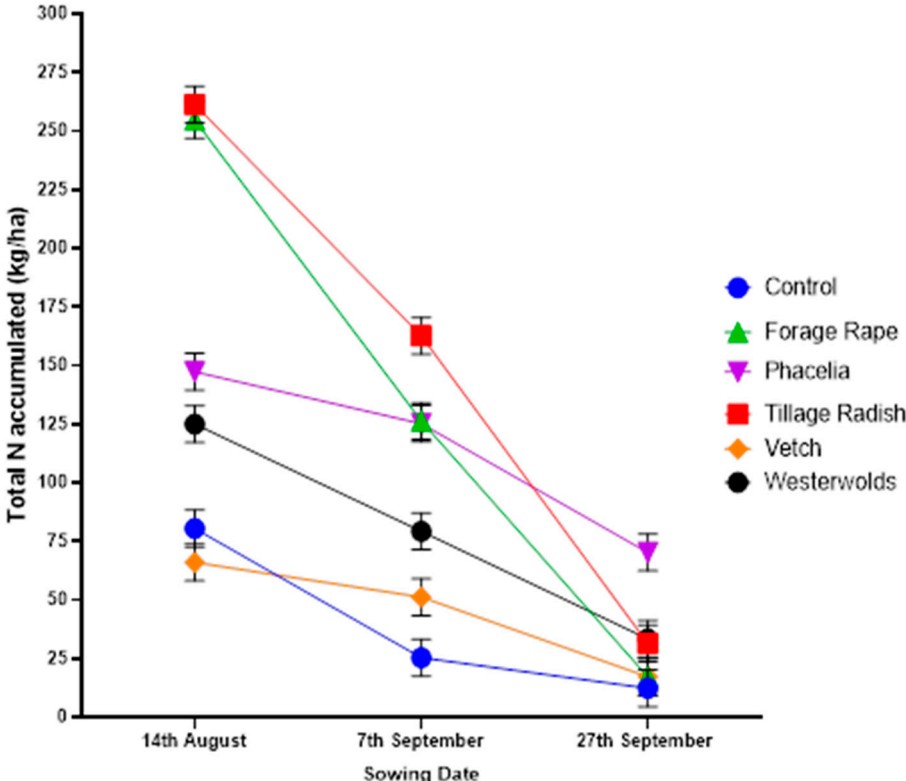

**Figure 3.** N accumulation for each treatment of cover crop which includes the above ground biomass, cover crop roots and weeds produced for each of the three sowing dates. Error bars represent the standard error of the mean (SEM). Fisher's unprotected $LSD_{0.05}$ = 41.0. N = 8 for each mean.

The control accumulated over 80 kg/ha of N at SD 1, 25 kg/ha on SD 2 and 12 kg/ha on SD 3. This reflects a clear reduction in weed growth with delayed sowing. The amount of N accumulated in the control at SD 1 is atypical, as normal practice would not allow these weeds to produce viable seeds. Alternatively, a herbicide or additional cultivation would be used to destroy them. However, at later sowing, the weeds did not mature and the amount of weed N was higher in the phacelia sown at date 3 than the total weed N of the relative SD 3 control (14.6 versus 12.3 kg/ha). This shows a potential complementary competitive effect.

Sowing date significantly affected root N uptake ($p < 0.001$). At SD 1, tillage radish accumulated 48 kg N/ha compared to forage rape at 37 kg/ha. At SD 2, this declined to 15 kg/ha and 18 kg N/ha, respectively, and at SD 3 a maximum of 5 kg N/ha was detected in tillage radish.

*3.7. Carbon (C) Accumulation*

C accumulation exhibited significant ($p < 0.001$) differences in sowing date, species and a significant interaction ($p < 0.001$). Tillage radish and forage rape accumulated the greatest total C of 2359 and 2361 kg/ha, respectively (Figure 4). Phacelia exhibited the greatest growth at SD 3, which resulted in the largest C accumulation (Figure 4). C accumulation in the roots was only affected by sowing date ($p < 0.001$) due to no significant difference in the species average concentration. The controls produced the least C at each sowing date.

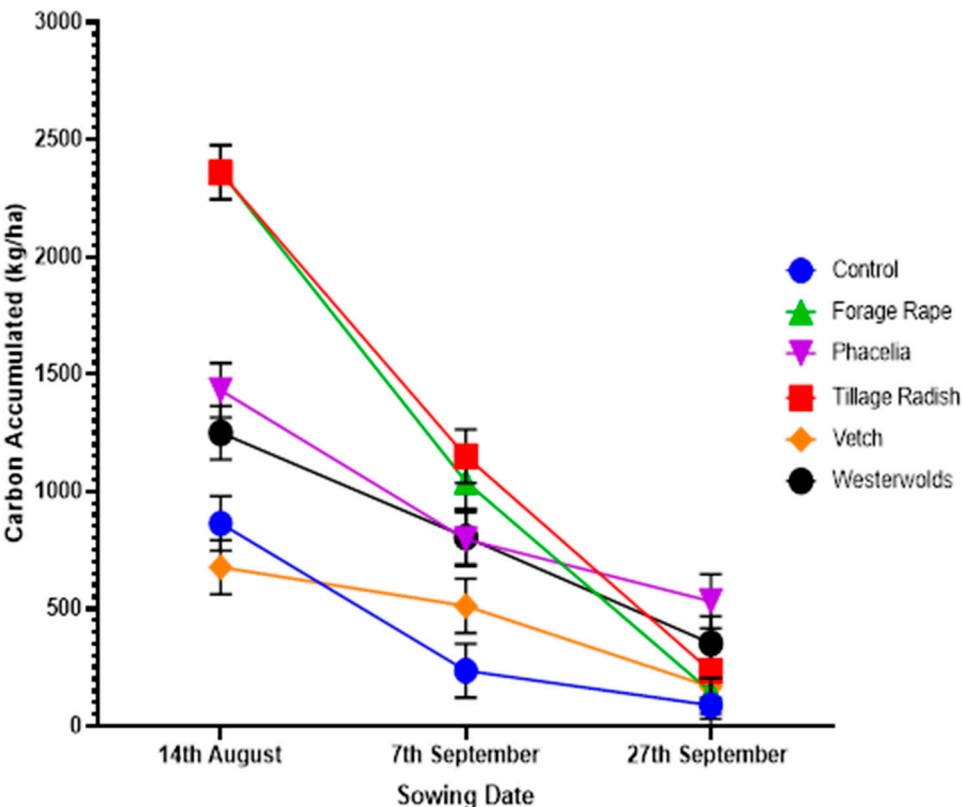

**Figure 4.** Carbon (C) accumulation for each treatment of cover crop which includes the above ground biomass, cover crop roots and weeds produced for each of the three sowing dates. Error bars represent the standard error of the mean (SEM). Fisher's unprotected $LSD_{0.05}$ = 327.7. N = 6 for each mean.

### 3.8. C:N Ratio

SD 1 had a significantly higher C:N ratio than SD 2 or SD 3 ($p < 0.05$) (Table 3). C:N ratio was significantly different ($p < 0.01$) between species, with the tillage radish having the lowest C:N ratio of all species ($p < 0.05$). The C:N ratio exhibited a low range varying between 8.0 to 10.7.

**Table 3.** C:N ratio of cover crop biomass at the different sowing dates (excludes roots).

| Sowing Date | Control | Forage Rape | Phacelia | Tillage Radish | Vetch | Westerwolds | Sowing Date Average |
|---|---|---|---|---|---|---|---|
| 1 | 10.6 | 8.9 | 10.8 | 9 | 8 | 10.4 | 9.6 [b] |
| 2 | 10.5 | 8 | 6.4 | 6.9 | 8.4 | 8 | 8.0 [a] |
| 3 | 11 | 9.1 | 7 | 8.1 | 8.5 | 9.3 | 8.8 [a] |
| Species Average | 10.7 [c] | 8.7 [ab] | 8.1 [ab] | 8.0 [a] | 8.3 [ab] | 9.2 [bc] | 8.8 |
| | | | | Statistical analysis | | | |
| | Treatment | | *p*-value | SEM * | LSD [#] | | |
| | Sowing Date | | <0.001 | 0.494 | 0.912 | | |
| | Species | | <0.01 | 0.946 | 1.285 | | |
| | Sowing Date × Species | | 0.06 | 0.489 | 2.205 | | |

* Standard error of the mean. [#] Least significant difference (0.05). Means which do not share the same letters are significantly ($p < 0.05$) different to each other. N = 6 for each mean.

### 3.9. Macro-Nutrient Assimilations (P, K, and S) of Cover Crops

Macro-nutrient assimilation values are presented in Figure 5, and whilst Table 4 shows the significance of treatments applied. Effects of sowing date and species exhibited significant ($p < 0.001$) differences for P, K, and S, and all interactions were significant ($p < 0.001$).

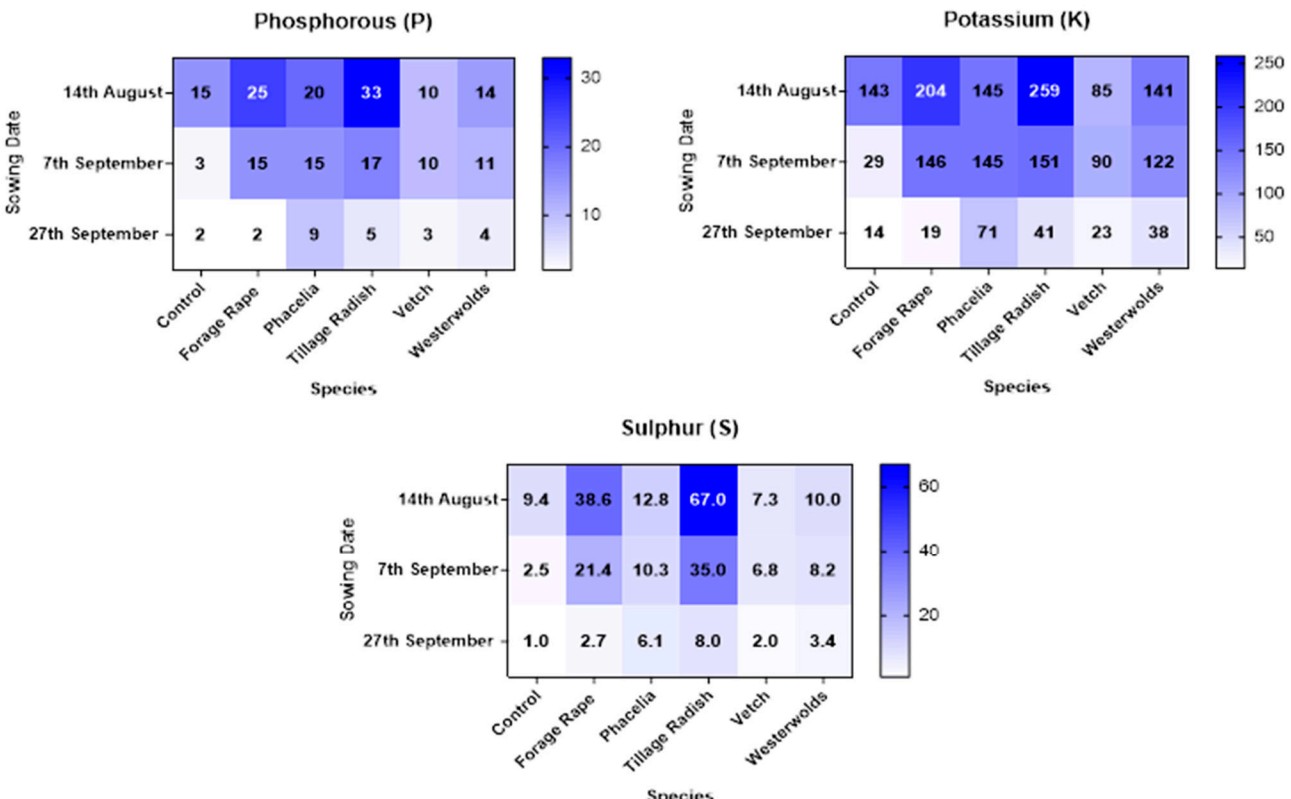

**Figure 5.** Nutrient assimilation of P, K and S of the cover crops in response to sowing dates (kg/ha).

**Table 4.** *p*-values for the nutrient assimilations for the treatments of sowing date, species and their interaction.

| | P Uptake (kg/ha) | | | K Uptake (kg/ha) | | | S Uptake (kg/ha) | | |
|---|---|---|---|---|---|---|---|---|---|
| | Chi-Prob | SEM * | LSD # | Chi-Prob | SEM | LSD | Chi-Prob | SEM | LSD |
| Sowing Date | *p* < 0.001 | 0.9 | 2.7 | *p* < 0.001 | 9.3 | 22.6 | *p* < 0.001 | 1.12 | 2.97 |
| Species | *p* < 0.001 | 1.0 | 3.8 | *p* < 0.001 | 9.3 | 32 | *p* < 0.001 | 1.19 | 4.14 |
| Sowing Date × Species | *p* < 0.001 | 0.9 | 6.6 | *p* < 0.001 | 9.0 | 55.4 | *p* < 0.001 | 1.07 | 7.16 |

* Standard error of the mean. # Least significant difference.

Tillage radish at SD 1 accumulated the greatest P (33 kg/ha), K (259 kg/ha), and S (67 kg/ha), demonstrating that this species has the largest potential to accumulate nutrients when sown early (Figure 5). The slurry added 50 kg/ha of P. In comparison to the control, the tillage radish accumulated almost two-fold more P, with the tillage radish accumulating 25 kg/ha P, which is half of what the slurry added. At SD 2, tillage radish, forage rape and phacelia accumulated 94%, 91% and 91%, respectively of the 160 kg/ha of K added by the slurry. However, at SD 1, tillage radish and forage rape accumulated 99 and 44 kg/ha more K than supplied by the slurry. The high concentration of S in the tillage radish and its large biomass resulted in an uptake of 67 kg/ha S being accumulated. This is three-fold greater S than added by the slurry. At SD 2, which represents sowing after a normal harvest of cereal crops, there is less variation in the uptake between tillage radish, forage rape and phacelia. However, at SD 3, phacelia accumulated the greatest levels of P, and K but not S.

### 3.10. Grain Yield

At harvest of the spring barley, many plots exhibited excessive chickweed growth due to partial resistance to the herbicides used. This could have affected grain yield. Therefore, the chickweed was visually scored prior to harvest on a plot basis, along with lodging, leaning and brackling. Grain yield was analysed using REML with the covariates of lodging, leaning and brackling to produce predicted mean grain yields. Chickweed was the only

covariate with a significant effect ($p < 0.001$) on grain yield. The REML analysis using the covariates found that grain yield was unaffected by any sowing date or species (Table 5) and yields varied between 6.9 and 7.9 t/ha (85% DM). Numerically, barley, following phacelia, exhibited the highest species average yield and the highest individual yield when sown at the latest sowing date. Westerwolds sown at SD 3 exhibited a 1 t/ha lower grain yield than phacelia. All sowing dates had similar mean grain yields.

**Table 5.** 2018/19 spring barley yield (t/ha) and REML analysis.

| Sowing Date | Control | Forage Rape | Phacelia | Tillage Radish | Vetch | Westerwolds | Sowing Date Average |
|---|---|---|---|---|---|---|---|
| 1 | 7.52 | 7.28 | 7.41 | 7.5 | 7.51 | 7.67 | **7.48** |
| 2 | 7.43 | 7.34 | 7.45 | 7.31 | 7.44 | 7.54 | **7.42** |
| 3 | 7.41 | 7.61 | 7.94 | 7.32 | 7.72 | 6.88 | **7.48** |
| **Species average** | **7.45** | **7.41** | **7.6** | **7.38** | **7.56** | **7.36** | **7.46** |
| | | | REML Analysis + covariates | | | | |
| | Parameter | | | *p*-value | SEM # | LSD * | |
| *Covariate* | *Chickweed* | | | **<0.001** | | | |
| *Covariate* | *Leaning* | | | 0.94 | | | |
| *Covariate* | *Lodging* | | | 0.42 | | | |
| *Covariate* | *Brackling* | | | 0.11 | | | |
| Variate | Sowing Date | | | 0.85 | 0.106 | 0.403 | |
| Variate | Species | | | 0.81 | 0.108 | 0.412 | |
| Variate | Sowing Date × Species | | | 0.47 | 0.104 | 0.778 | |

# Standard error of the mean. * Least significant difference. N = 4 for each mean.

## 4. Discussion

### 4.1. Grain Yield

The success of cover crops can be judged on the grain yields produced, due to its economic importance. However, the grain yield was not significantly affected by any of the treatments or combinations in this single year study, which would question the value of using cover crops. The grain yields exhibited are all regarded as very high for spring barley [16,17]. A second year's replication of the trial was attempted but adverse weather of high rainfall during the planned sowing dates meant that it was impossible to initiate because the soil was not trafficable.

The lack of difference in grain yield may be due to an oversupply of N which masked effects between treatments. Prior to the planting of the cover crops (August 2018), an average of 60 kg/ha of inorganic N was found in the 15 cm soil profile and 263 kg N/ha (total N) supplied from slurry and the additional 70 kg N/ha in the form of inorganic fertiliser. This meant that N could have been oversupplied causing the lack of differences. In this trial, winter rainfall was low (Figure A3) and could have retained more N in the soil and from the slurry. Table A5 shows the SMN of control plots (fallow) sampled on 25 February 2019 post-winter and prior to flailing. This ranges from 9.5 to 24.4 kg N/ha (15 cm profile), which shows a considerable decline between the initial (August) SMN and the post-winter (February) measurement. This decline could be a combination of leaching/loss of N or immobilisation into the soil. Based on the February assessment of SMN and using a typical estimation for N availability from slurry, was why an extra an additional 70 kg N/ha was applied to the spring barley. A study by White et al. [18] investigating a range of applications of N applied in the autumn found that, in NI, SMN tested in the spring was not a good predictor of soil N supply. Retrospectively, this should not have been applied (in this year). This may have been due to sufficient rainfall and soil temperatures that promoted a high rate of N mineralisation in the soil which, in turn, supported the N requirement of the spring barley. However, White et al. [18] found that application of 800 kg N/ha in the autumn in NI still required additional N to be applied in the spring/summer growing season to maximise the yield of wheat as a result of the N losses and immobilisation.

The different cover crop residue quantities, and qualities, can add variability to nutrient supply of the commercial crop [10] as increasing C supply to soil microorganisms can immobilise nutrients in the microbial biomass and compete with roots for N [19,20]. This

can reduce commercial crop yield [21]. This trial suggests two reasons why cover crops are not immobilising N in the conditions applied, and that the residue is mineralising at a sufficient rate to supply the spring barley with adequate nutrients.

1. No significant increase in yield was found, in any treatment, in response to the 70 kg/ha of inorganic N applied.

2. The sowing dates produced various quantities of biomass, whereby increased biomass could have increased immobilisation.

Slurry was applied prior to sowing to ensure that nutrients were not a limiting factor, in order to maximise cover crop biomass, and thus identify nutrient uptake. The rationale of the treatment design implemented, e.g., applying slurry, was that it was thought that the cover crops were going to have low N mineralisation rates with potential N immobilisation, particularly the brassica species as found by Couëdel et al. [11]. Findings from this experiment and Cottney et al. [13] show a considerable breakdown of nutrients from the cover crops, with the ability to replace all inorganic N required by spring crops. In this trial, N uptake in the cover crops varied from 20 to 261 kg/ha, whereas the control had 17 kg N/ha on average in February. Assuming grain N of 1.5% and that the grain accounts for 80% of N accumulated, a spring barley crop with a grain yield of 7 t/ha and 8 t/ha (15% moisture content) would require 111 and 127 kg N/ha, respectively. If SMN from the February control (only 15 cm) plots' average is subtracted, this means the barley requires an estimated 94 and 110 kg N/ha. This suggests that the spring barley acquired N from sources other than the fertiliser and residue, especially in the control plots of SD 3 which accumulated 12 kg N/ha in the weed residue. Therefore, the 70 kg of inorganic N should not have been applied to the spring barley. However, verification of this decision would have required a treatment of plots with a zero rate N fertiliser regime as well as the 70 kg/ha of inorganic N. Increasing the reliability of these results would require more replication across different years and particularly at different sites, as this was a site of high fertility.

### 4.2. Biomass Yield

To maximise the area sown to cover crops requires species which produce adequate growth when sown late. This gives farmers the assurance that they are not wasting their resources. This research demonstrates that, at SD 3, phacelia produced the greatest biomass and was the best-suited species in that slot. At SD 1, a maximum biomass growth of 6447 kg/ha DM was recorded from tillage radish, which reduced to 307 kg/ha DM following the final SD. When cover crops are grown as a livestock feed, the cost of forage is proportional to the level of growth and utilisation of that crop, thus favoring high biomass to dilute costs and make the crop profitable. The quantity of biomass produced from the forage rape was 6062, 3346 and 307 kg/ha following SD 1, 2 and 3, which is high in comparison to a trial conducted in Ireland by Keogh et al. [3]. They found that the maximum dry matter yield of forage rape (cv. Stego) when sown on the 1 August was 4548 kg DM (shoot + root DM), declining to 3047 kg when planted on 15 August and 1091 kg DM on 31 August. All had been supplemented with 120 kg N/ha. The other species investigated in that experiment was stubble turnips (cv. Delilah) which exhibited larger biomass yields at the latter two sowing dates. Furthermore, only the late-sown (SD 3) forage rape was damaged considerably by pigeons. It is, therefore, unsuitable when sown late. This phenomenon was only observed in late-sown plots and is presumed to be due to a combination of production of a dense canopy when sown at date 1 and 2. Consequently, pigeons could not land to graze.

### 4.3. N Accumulation

Tillage radish accumulated 261 kg N/ha at SD 1, which is almost twice the N requirement for a 6 t/ha spring barley crop [22]. This large accumulation was driven by high biomass and high% N. Phacelia, forage rape and tillage radish contained over 4% N, whereas a study by Wendling et al. [23] found that phacelia, tillage radish (daikon)

and turnip rape (forage rape) had a N concentration of 2.13, 2.22 and 2.04%, respectively. The higher% N in this trial reflects the high levels of N contained in the soil this trial was conducted on. Moreover, these cover crops can increase concentration of N in response to additional nutrients (slurry) [13]. This is a mechanism to increase N uptake and reduce the C:N ratio. Wendling et al. [23] reported biomass yields of 6.3, 6.3 and 4.4 t/ha and N uptakes of 120, 139 and 132 kg/ha for phacelia, tillage radish and forage rape, respectively. In this trial, tillage radish, phacelia and forage rape all accumulated considerably higher quantities of nutrients despite having similar biomass. This is due to the considerably greater concentration of N. This study found that vetch had the greatest N concentration when planted early (5.16%), which declined in response to delaying sowing to 2.94 and 3.09% at SD 2 and SD 3, respectively. In comparison, Lawson et al. [8] found that delayed sowing did not affect N concentration in the hairy (winter) vetch (variety not stated) and in comparison,% N averaged 4.1% and the monoculture was 31% weeds with a biomass of 1.4 t/ha. The species' average% N was considerably higher than many other studies [8,24]. This may be due to high residual N with the species increasing their relative N concentration in response to the additional nutrients in slurry, thus creating a luxury uptake.

The vetch, a legume, should have fixed additional N into the rhizosphere. The process and productivity of N fixation is not only temperature-dependent but also relies on bacterial infection of the roots. This occurs 3–4 days post-germination and takes 3–5 weeks to produce visible and active root nodules [25], which means that delayed sowing would highly effect the vetch's ability to biologically fix N. Li et al. [26] estimated that legumes fix 24 kg of N per tonne of biomass produced. Extrapolating those findings means that in this study only moderate levels of N could have been fixed, as only 242, 638 and 189 kg/ha of biomass was produced from SD 1, SD 2 and SD 3, respectively. However, this study underestimates the N contained in the roots as they were not considered. This has been found to equate to 30–50% of plant N [26].

The N in the taproots of the brassicas was evaluated but, again, it does not account for minor roots as well as N rhizodeposits through sloughed-off root hair cells, N in root exudates and root fragments which have been found to account for an additional 4.6–10.3% of total plant N for brassicas (tillage radish, winter turnip rape and oilseed radish [27]. This means that the N accumulations have been underestimated due to difficulty in accurately extracting these N rhizodeposits under field conditions. Furthermore, at SD 1, there will be a greater proportion of N rhizodeposits because the biomass in the roots declined with delaying sowing, as seen in Figure 2.

Tillage radish accumulated 261 kg N/ha, which is considerably more N than any spring commercial crop requires. Furthermore, it also contained 260 kg of K and 67 kg of S and is thus a considerable bio-fertiliser. However, this uptake of macro-nutrients could diminish soil nutrient availability in the spring and affect commercial crop yields, but would depend on nutrient mineralisation rate. Therefore, subsequent trials must investigate immediate impact on soil fertility. The cultivated control accumulated 80 kg N/ha in the weeds at SD 1 and is arguably not representative of farmer practice as it would have been destroyed with herbicides to avoid weeds producing viable seeds. Therefore, it is an overestimate of what typically fallow land would have accumulated. The Nutrient Management Guide RB209 [22] estimates soil residual N following harvest of various crops and rotations, where a SNS of 70 kg/ha would be relatively high. This means that phacelia sown late (accumulating 70 kg/ha at SD 3) has the potential to deplete these SNS reserves and could be beneficial to mitigate against N leaching. Therefore, the functions of cover crops at later sowing dates would change to more predominantly environmental considerations of N accumulation to reduce leaching and physical soil protection, due to numerous other benefits being linked to biomass production, such as effect on soil biology [28].

The decline in both biomass and N uptake of the roots at later sowing of the tillage radish and forage rape suggests that this might reduce their ability to "biodrill"—a term referring to the ability of roots to grow through a plough pan (layer of compacted soil) to

enhance soil structure [29]. This is due to the number of roots being similar, since the same number of seeds were planted, although individual root size has been reduced. This could be detrimental to growing deep or exerting a positive effect on the soil profile, as thicker roots can exert higher penetrative pressure [30,31].

### 4.4. C:N Ratio

The% N of species in this experiment was high. This led to a low C:N ratio, especially in comparison to other studies. For example, forage rape had a C:N ratio of 22 when sown as a sole crop by Couëdel et al. [11], where tillage radish (same variety) and phacelia had a C:N ratio of 18.6 and 20.6 reported by Wendling et al. [23]. In this trial, the highest C:N ratio was 10.0 in the control, with all cover crop species having lower C:N ratios. This suggests that the plant nutrients will break down quickly, leading to a net mineralisation [12]. This may have been observed, as the different sowing dates producing various quantities of biomass and thus N uptake did not negatively influence spring barley yield. In subsequent trials, the N offtake of spring barley must be evaluated to help identify effects. Couëdel et al. [11] found that N mineralisation to the commercial crop ranged from −6 kg/ha for mustard to 20 kg/ha in the forage rape (same variety). Silgram and Harrison [10] estimated that cover crops with a C:N ratio below 25–30 are required for net mineralisation in year one. However, Couëdel et al. [11] concluded that a threshold of below 15 was required for net mineralisation within 6 months.

The cover crops in this research were destroyed as early as weather permitted and, if delayed to a later date, C:N ratio would have increased due to greater amounts of structural compounds within the plant. This could reduce the rate of release of nutrients from the residue, and decrease the transfer of nutrients to the following commercial crop, and thus increase the N requirement for that crop [32]. The C:N can be modified by species choice and supply of supplementary nutrients in the form of slurry, as found by Cottney et al. [13], meaning that nutrient mineralisation can be manipulated. Jensen et al. [33] found that N released from plant residues after 217 days reached a maximum of 40% for those with a C:N below 10. When applied to this study, 104 kg of N would have been released from the tillage radish on SD 1.

### 4.5. C Accumulation

Increasing soil organic matter levels is beneficial not only for soil functions including porosity, biological activity, nutrient retention and soil structure but can also to help offset anthropogenic carbon dioxide emissions (CO2) [34]. This study shows that early sowing is paramount for returning C. Forage rape returned over 2361 kg C on SD 1 equivalent to 8.66 t/ha $CO^2$ [35] despite forage rape producing less biomass than the tillage radish. Forage rape had a higher% C than tillage radish which resulted in the larger C accumulation. In comparison, the control accumulated the lowest C in biomass. This demonstrates the environmental benefit of cover crops compared to fallow. Alternative ways to return organic material to soil is to incorporate the commercial crop straw but this comes at a cost when this material is a commodity. Average straw yields for spring barley in 2017 and 2018 was 3.5 t/ha and 5 t/ha at 46% C (determined in a prior experiment) which means a return of 1.6 t/ha and 2.3 t/ha of C. This organic material will have a higher C:N ratio and will be slower to decompose, which may be better to improve long-term organic matters [36,37]. However, the typically high straw prices in regions such as NI contribute considerably to gross margins [38], whereby incorporating straw as a method to return organic matter is not justified financially/economically and that cover crops shown are just as effective at returning similar levels of C to the soil. This study does not account for additional sources of C that the cover crops are returning which originate from root exudates. They can be a significant proportion of photosynthetic C transformed into plant and microbiome usable products as Swinnen et al. [39] found in wheat and barley that rhizodeposited C represented 7–15% of total C assimilated.

### 4.6. P, K, and S Uptake of Cover Crops

The slurry applied 50 kg/ha P (90% availability) [22], and tillage radish accumulated 33 kg/ha of P at SD 1. This is a substantial uptake of the added nutrients which could have considerable environmental benefits. In comparison, phacelia at SD 3 accumulated 9 kg/ha P, and had the greatest accumulation at this date. Whilst nutrient loss from groundwater was not measured, trapping nutrients in biomass is a mechanism of protection against loss from the system [6]. Furthermore, on low nutrient index soils, the cover crops could sequester nutrients and cause a negative effect on the growth of the commercial crop due to nutrient competition. Couëdel et al. [40] found a maximum S uptake of 23 kg/ha for tillage radish whilst forage rape (mosa) had an uptake of 17 kg/ha, whereas this study found that the same species exhibited almost a three-fold greater uptake. Slurry supplied 24 kg/ha of S, which suggests that tillage radish can effectively sequester the S applied. Early-sown tillage radish and forage rape accumulated large amounts of K which could either enhance nutrient cycling if this mineralises quickly, or could cause competition for K in the spring barley crop. Slurry added 160 kg/ha K, whereas tillage radish accumulated 260 kg/ha thereby facilitating the fulfilment of the cover crop requirement and replenishing the soil. Soil tests post-cover crop or commercial crop were not conducted but investigating this could be important to determine whether cover crops facilitate nutrient cycling.

### 4.7. Weed Suppression

Weed pressure decreased naturally with later sowing and was observed through diminished biomass in the control with later planting. At SD 1 and SD 2, tillage, forage rape and brassicas exhibited almost total weed suppression. This demonstrates that these species, and in particular, forage rape could be a viable alternative to the chemical control of weeds. Forage rape at SD 1 and SD 2 was unaffected by the frost, retaining its canopy and was thus able to shade weeds and provide almost total weed suppression. This is observed in the final ceptometer reading, whereby forage rape had the highest LAI in comparison to the other species. The weed suppression in phacelia and tillage radish was high due to their profuse growth, also found by Brust et al. [9]. LAI was only measured on one date for SD 3 for two reasons. Firstly, the ceptometer uses two different sensors to measure incoming radiation. This can create an error which increases in plots with sparse biomass. Secondly, it was not possible to measure very low biomass as the wand could not get under its canopy, creating a null measurement.

If planting for weed suppression, the best cover crops are tillage radish, phacelia, forage rape and westerwolds. This is of particular importance in organic systems. However, weeds at SD 3 could be regarded as beneficial to trap more nutrients and add to soil structure protection due to growing roots anchoring the soil.

### 4.8. Recommendations to Maximise and Encourage Later Sowing of Cover Crops

Cottney et al. [41] found that a lower proportion of farmers considered planting cover crops after commercial crops harvested in September. This reduces the amount of land sown to cover crops. By planting later, through using better suited species as identified in this study, would reduce the amount of land left fallow over winter and could mitigate against loss of nutrients such as N, and also provide many more benefits. This study has found that species choice when sowing later is critical. Another strategy to improve species competitiveness when late sown, is to increase seed rate. This was not investigated in this trial but could be implemented on farm to enhance growth. Other strategies include using a mixture of species to encourage competition (also referred to as over-yielding) which was demonstrated in a study by Wendling et al. [42]. It was also found in this trial that the N fractions in the late-sown phacelia contained more weed N than the relative control covered in weeds. However, where supplementary N is applied, increased competition from a mixture is diminished in comparison to the sole species alone [42]. Mixtures were not investigated in this trial due to the exponential number of combinations and seeding rates, whereas testing the individual species is of greater importance. This study recommends,

that if using a mixture of species or sole crop (one species) in planned rotations where harvest dates are after the start of September, to ensure that phacelia makes up a large proportion of that mix, or is the sole crop to be used.

Another strategy to encourage later sowing is to adopt subsidies. In the Republic of Ireland, with similar climatic conditions, which subsidises the practice of cover cropping, growers were more likely to plant after later-sown cereal rotations such as winter wheat and spring barley compared to those in NI [41]. The aims of the farmers, and species used, changed from being focused on grazing by livestock to a primary focus on soil structure, soil health and capture leachable nutrients. Therefore, subsidies could be adopted in NI, especially as this trial has demonstrated that cover crops can capture considerable nutrients in these climatic conditions.

## 5. Conclusions

Spring barley yield was unaffected by later sowing of cover crop species. Phacelia competed considerably better than all other species when late sown, accumulating 70 kg/ha of N with a considerable biomass. This is a large amount of N that is protected over winter in comparison to fallow. Furthermore, the N accumulation by phacelia is almost half the requirement of a subsequent spring barley crop, if the N in the biomass mineralises sufficiently. Early-sown tillage radish and forage rape accumulated over 250 kg N/ha, which is almost twice what spring crops require. This further highlights how unproductive fallow land is and that modern agriculture cannot close the nutrient cycle status quo.

**Author Contributions:** Conceptualization, P.C., P.W., L.B. and E.W.; Investigation, P.C.; Writing—Original Draft, P.C.; Writing—Review and Editing, P.C., P.W., E.W. and L.B.; Visualization, P.C.; Resources, L.B. and E.W.; Supervision, P.W., L.B. and E.W. All authors have read and agreed to the published version of the manuscript.

**Funding:** This research was funded by DAERA grant number (17/1/01).

**Institutional Review Board Statement:** Not applicable.

**Informed Consent Statement:** Not applicable.

**Data Availability Statement:** Not applicable.

**Acknowledgments:** The authors would like to thank all AFBI staff involved in helping undertake this trial, in particular, Colin Garrett Ashley Cathcart and Gary Weir as well as all the farm staff who helped with setting up the trial. Seed was kindly provided by RAGT Seeds. The authors are also grateful to the AFBI statistical analysis branch and acknowledge the guidance and analysis conducted by Michelle Allen (AFBI, Newforge), and likewise, Hugh McKeating and Alan Wright (AFBI, Newforge) of the soils branch for analysing soil and plant samples.

**Conflicts of Interest:** The authors declare no conflict of interest. The funders had no role in the design of the study; in the collection, analyses, or interpretation of data; in the writing of the manuscript, or in the decision to publish the results.

## Appendix A

**Table A1.** Cover crop, varieties, sowing rates and plants sown/m$^2$.

| Species | Variety | Recommended Sowing Rate (kg/ha) | Chosen Sowing Rate (kg/ha) | Plants Sown/m$^2$ |
|---|---|---|---|---|
| Tillage Radish | Daikon | 25–30 | 25 | 177 |
| Forage rape | Mosa | 10 | 10 | 278 |
| Phacelia | Natra | 10 | 8 | 462 |
| Vetch (Hairy) | Villana | 25–30 | 25 | 85 |
| Westerwolds | Magnum | 40–46 | 40 | 763 |
| Control/fallow | - | - | - | - |

**Table A2.** Nutrient concentration of slurry on a fresh basis and quantity (kg/ha) of nutrients applied.

| Parameter | K | P | S | Mg | NH$_4^+$ | Total N | DM |
|---|---|---|---|---|---|---|---|
| | mg/kg | mg/kg | mg/kg | mg/kg | % | % | % |
| | 4583.5 | 1438.5 | 671 | 874 | 0.5 | 0.8 | 7.8 |
| **Nutrient application (kg/ha)** | | | | | | | |
| Rate 35 m$^3$/ha | 160.4 | 50.4 | 23.5 | 30.6 | 176.2 | 263.7 | |

Nutrient concentrations are reported on a fresh basis.

**Table A3.** 2018/19 Spring barley agronomy and crop protection.

| Date | Active Ingredient | Chemical | Reason | Rate | Manufacturer |
|---|---|---|---|---|---|
| 28 May 2019 | Lambda-cyhalothrin | Warrior | Aphids | 50 mL/ha | Syngenta |
| | Manganese 500 g/L | Mantrac | Manganese Trace Elements | 1 lt/ha | YARA |
| | Chlorothalonil | Bravo | Disease | 1 lt/ha | Syngenta |
| | Mecoprop-P | Headland Charge | Weeds | 2 lt/ha | Headland Agrochemicals Limited |
| 6 June 2019 | Metsulfuron-methy + tribenuron-methy | Ally Max ® SX | Weeds | 42 g/ha | Dupont |
| | Prothioconazole + bixafen | Siltra Xpro | Disease | 0.6 lt/ha | Bayer |
| | Trinexapac-ethyl | Moddus | Growth—Regulator | 0.2 lt/ha | Syngenta |
| | Chlormequat chloride | 3C Chlormequat 750 | Growth—Regulator | 1 lt/ha | O-BASF |
| | Manganese 500 g/L | Mantrac | Manganese Trace Eements | 0.6 lt/ha | YARA |
| 2 July 2019 | Prothioconazole + bixafen | Siltra Xpro | Disease | 0.6 lt/ha | Bayer |
| | Chlorothalonil | Bravo | Disease | 1 lt/ha | Syngenta |
| **Fertiliser application** | | | | | |
| Date | Rate | Product | Manufacturer | Nitrate—N | Ammoniacal—N |
| 23 May 2019 | 70 Kg N/ha | Yara Can (27%) | YARA | 13.5% | 13.5% |

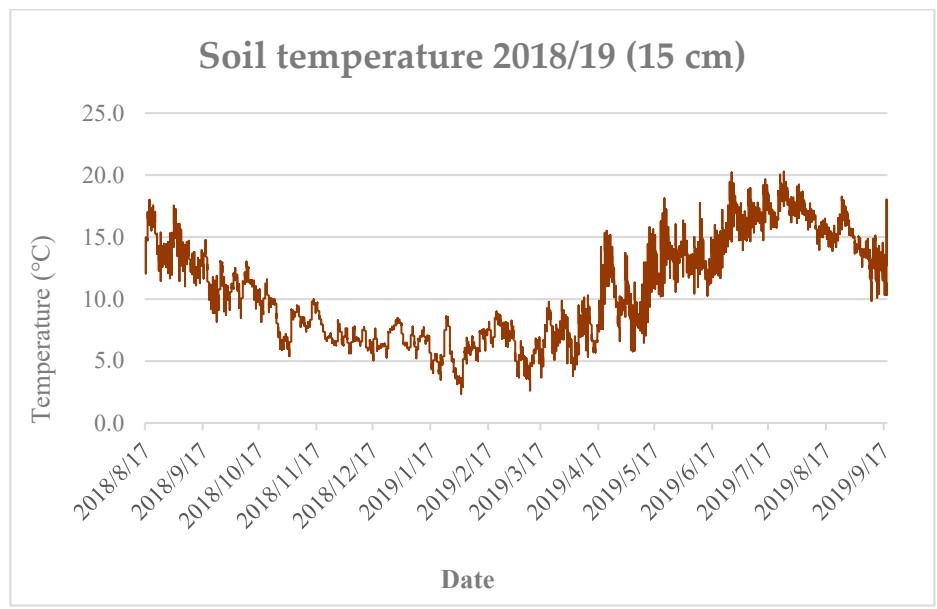

**Figure A1.** Recorded soil temperatures during both the cover crop and spring barley growth (°C).

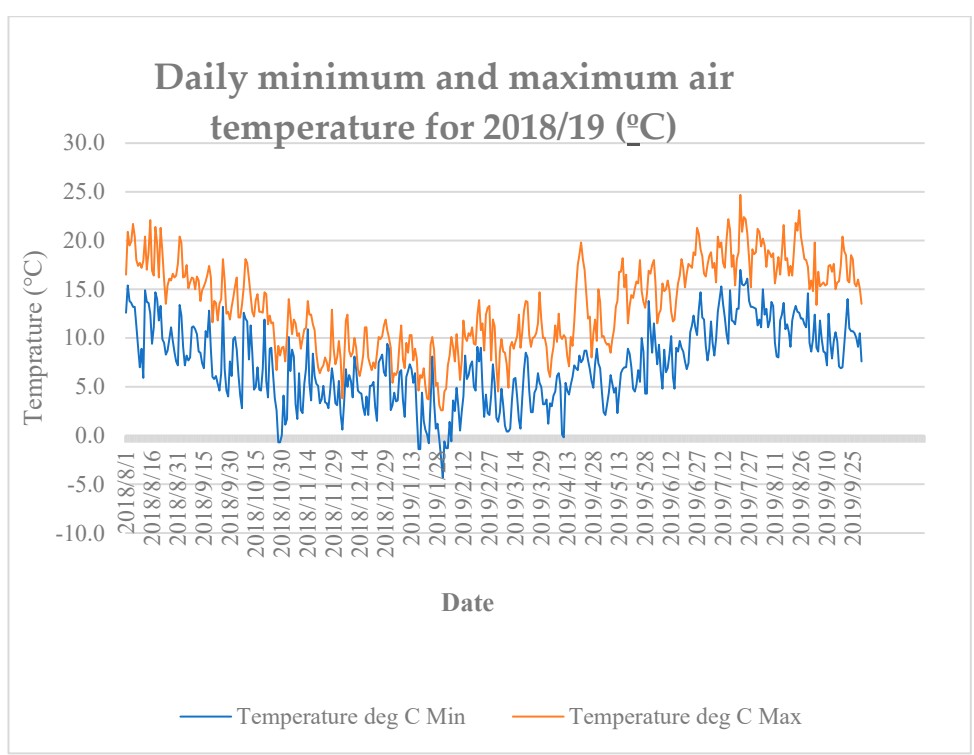

**Figure A2.** Recorded daily maximum and minimum air temperatures during both the cover crop and spring barley growth (°C).

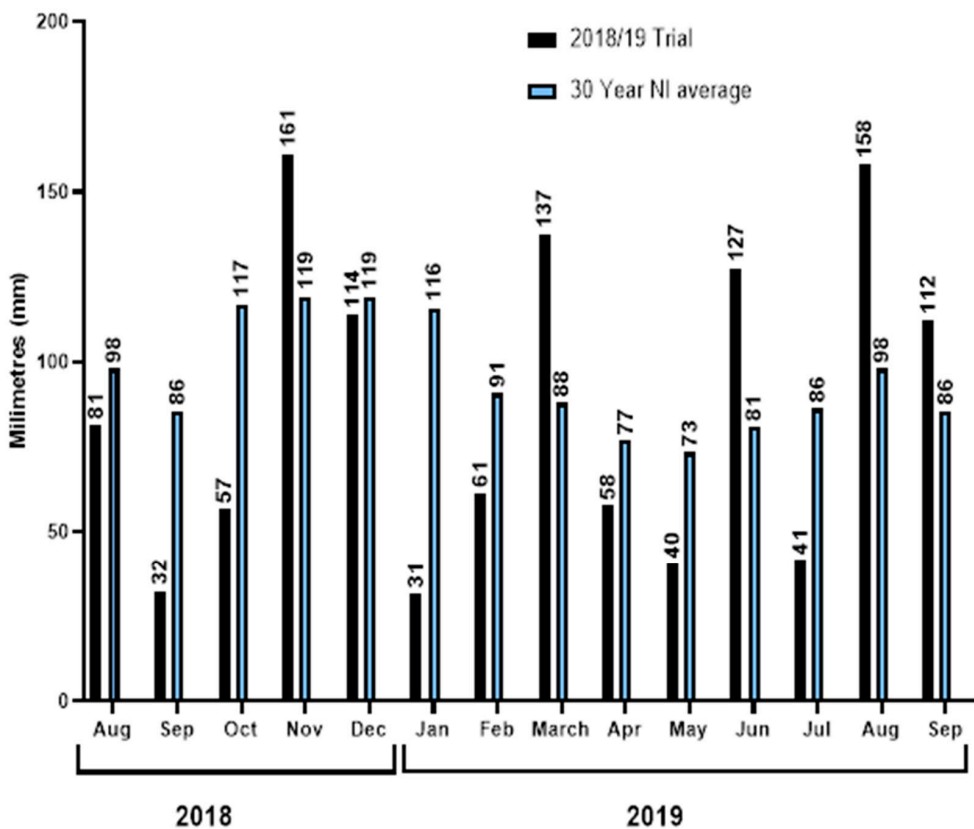

**Figure A3.** Recorded rainfall during both the cover crop and spring barley growth (mm). Recorded from an official weather station located Crossnacreevy, Belfast. The 30 year NI average was obtained from the Met Office.

**Table A4.** REML analysis *p*-values for parameters of cover crop growth.

| Parameter | Total CC * BIOMASS (kg/ha) | Total Root (R) Biomass (kg/ha) | Total Weed (W) Biomass (kg/ha) | Above Ground Biomass (CC + W) (kg/ha) | Total Biomass (CC + R+ W) (kg/ha) | % N CC (%) | % Carbon CC (%) | C:N Ratio CC | % N Roots (%) |
|---|---|---|---|---|---|---|---|---|---|
| Sowing date (SD) | <0.001 | <0.001 | <0.05 | <0.001 | <0.001 | 0.12 | <0.001 | <0.001 | <0.001 |
| Species | <0.001 | 0.64 | <0.001 | <0.001 | <0.001 | <0.001 | <0.001 | <0.01 | <0.05 |
| SD × Species | <0.001 | 0.52 | <0.001 | <0.001 | <0.001 | <0.01 | 0.08 | 0.06 | <0.05 |
| Parameter | % Carbon roots (%) * | Root C:N ratio | CC N uptake (kg/ha) | Root N uptake (kg/ha) | Weed N uptake (kg/ha) | Total N uptake (CC + root + weed) (kg/ha) | CC carbon accumulation (kg/ha) | Root carbon accumulation(kg/ha) | Total carbon (CC + root + weed) accumulation (kg/ha) |
| Sowing date (SD) | <0.05 | <0.001 | <0.001 | <0.001 | 0.35 | <0.001 | <0.001 | <0.001 | <0.001 |
| Species | <0.001 | <0.001 | <0.001 | 0.18 | <0.001 | <0.001 | <0.001 | 0.58 | <0.001 |
| SD × Species | 0.87 | 0.20 | <0.001 | 0.35 | 0.07 | <0.001 | <0.001 | 0.46 | <0.001 |

* CC—cover crop.

**Table A5.** Soil mineral nitrogen (SMN) of control plots measured on 29 February 2019.

| Sowing Date | Average SMN in kg/ha (15 cm) |
|---|---|
| 1 | 24.4 |
| 2 | 9.5 |
| 3 | 16.1 |
| Mean | 16.7 |

N = 4 for each mean.

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
