# Peer review of "How Cover Crop Sowing Date Impacts upon Their Growth, Nutrient Assimilation and the Yield of the Subsequent Commercial Crop"

_agronomy, doi:10.3390/agronomy12020369_

Round 1

Reviewer 1 Report

An interesting article giving a comprehensive analysis of the problem of cover crops in Northern Ireland. The influence of the date of sowing and the type of crop on the quantity and quality of cover crop biomass, weed biomass and subsequent commercial crop is estimated. It is of practical importance to determine the optimal cover crop for each sowing period. Interestingly, the revealed absence of a positive effect of cover crops on the yield of the main crop and the advantage of non-leguminous cover crops with a high level of nitrogen fertilizers. It would be interesting to see this experience for organic farming.

There are several data presentation issues that make it difficult to evaluate the results.

Lines 399-400: "When readjusted for chickweed” - it is necessary to present the initial yield, lodging, leaning, necking, brackling and chickweed scores. It is unclear how the yield was readjusted for Table 5 values.

Lines 95-96: “The trial site was located in Hillsborough, Co. Down, NI (Latitude 36.074600, Longitude -79.096352).” – this Hillsborough is placed in the USA, correct coordinates please.

Lines 178-179: “The nutrient accumulation of the weeds was not added on to the results shown” - why were weeds taken into account when determining nitrogen and carbon, but not when determining nutrient assimilations? It would be logical to take weeds into account.

Lines 182-184: “Restricted maximum likelihood (REML) was used to analyse the nutrient accumulation due to the unequal number of observations." - it is also necessary to explain why REML was used to analyze LAI, barley yield with an equal number of observations.

Line 211-213 Figure 1: Why is SED shown? It would be logical to show SEM, as in Figures 2-4.

Lines 258-259: "Overall weed growth (measured from control plots) was 2744 kg/ha, 2271 kg/ha and 717 kg/ha at SD 1, SD 2 and SD 3, respectively” - in Fig. 2, SD2 control is clearly less than 2000 kg/ha. Why are the control strips green, despite the fact that weeds should be marked in red?

The remaining comments are editorial.

Lines 27-28: “Keywords: cover crops; nitrogen assimilation; spring barley yield; weed management; carbon assimilation; biofertiliser; light interception carbon assimilation

Lines 46-47: “can cover crops be can be sown and grow “

Line 75: "nutrient cycling and grain yield in that pot experiment” - why “pot experiment”?

Line 80: “The objective is to investigate the effect on;" - you need a colon instead of a semicolon at the end.

Line 114: ”Sowing dates (SD) were 14th August 2018 (SD 1), 7th September 2018 (SD 2), and 113 27th September 2018 (SD 3), respectively

Line 181: “Genstat Version 18 (VSN VSN International, 2017)”

Lines 187-189: "Results are deemed significant if probability due to random chance is under 5% (P<0.05) and trends are considered under 10% (P=0.10)”" - it would be correct to write “P<0.10”

Line 211-213: “Figure 1. Ceptometer readings showing the leaf area index (LAI) of the three different sowing dates for each cover crop measured at the monthly dates. Error bars represent standard error of the difference (SED). Steep declines in LAI are due to natural senescence and frost. Early sown crops were more affected by frosts than later sowing. N = 8 for each mean" – a lot of unnecessary information.

Line 217: "Table 2. REML analysis of ceptometer readings" - it would be better to write “Table 2. REML analysis of leaf area index”

Figure 2-4: "least significant difference (LSD) 0.05 = 884.7”. - mathematically incorrect “0.05=884.7". It would be better to write everywhere LSD0.05=884.7.

Line 325: Figure 3 – "Error bars represent standard error of the mean (SEM) (7.89)

Lines 357-358: “(SEM) (115.3).”

Lines 391-392: “Figure 5. Nutrient assimilation of P, K and S of the cover crops in response to sowing dates (kg/ha) Slurry supplied 50 391 kg/ha P, 160 kg/ha K and 24 kg/ha S

Line 646: "4.8. Recommendations to maximize and encourage later sowing of cover crops” - it is necessary to highlight the sentence as the subtitle.

Line 693 Table A1. TGW 2019/20 - this experience is not in the manuscript, its redundant data.

Line 704: “Table A3. 4.6 2018/19 Spring barley …”

Line 736: Figure A.1 Recorded soil temperatures during both the cover crop and spring barley growth (°C - there is no closing parenthesis. Long-term average values would be interesting, as well as for precipitation on Figure A.2.

Line 760: Figure A2 - it is necessary to specify that this is the air temperature.

Line 792: Figure A.2 is the wrong number, this is Figure A3. The name of the vertical axis "Millimeters (mm)" is incorrect, the correct one is "precipitation (mm)".

Table A.5 - in column 1, instead of SD1, SD2, SD3, it would be logical to write just 1, 2, 3, as in Tables 3 and 5.

The list of references is not designed according to the rules of the journal.

Author Response

Dear reviewer thank you for taking the time to read over the manuscript and provide feedback. This feedback has been acted upon with extensive changes highlighted both in the manuscript using track changes along with a point by point response to each comment. The point by point response to the comments is in blue writing. Furthermore, sections from the manuscript have been inserted into the response to comments with the changes made in red text whilst original text in black.

An interesting article giving a comprehensive analysis of the problem of cover crops in Northern Ireland. The influence of the date of sowing and the type of crop on the quantity and quality of cover crop biomass, weed biomass and subsequent commercial crop is estimated. It is of practical importance to determine the optimal cover crop for each sowing period. Interestingly, the revealed absence of a positive effect of cover crops on the yield of the main crop and the advantage of non-leguminous cover crops with a high level of nitrogen fertilizers. It would be interesting to see this experience for organic farming.

There are several data presentation issues that make it difficult to evaluate the results.

Lines 399-400: "When readjusted for chickweed” - it is necessary to present the initial yield, lodging, leaning, necking, brackling and chickweed scores. It is unclear how the yield was readjusted for Table 5 values.

Wording is incorrect and not is not a readjustment rather REML with covariance analysis was used to predict mean grain yields using the covariates described in the methods. Please see amendment to text in section 3.11 grain yield and the text is also shown below in red for your convenience

At harvest of the spring barley many plots exhibited excessive chickweed growth due to partial resistance to the herbicides used. This could have affected grain yield. Therefore, the chickweed was visually scored prior to harvest on a plot basis, along with lodging, leaning and brackling. Grain yield was analysed using REML with the covariates of lodging, leaning and brackling to produce predicted mean grain yields. Chickweed was the only covariate with a significant effect (P<0.001) on grain yield. The REML analysis using the covariates found that grain yield was unaffected by any sowing date or species (Table 5) and yields varied between 6.9 and 7.9 t/ha (85% DM). Numerically, barley following phacelia exhibited the highest species average yield and the highest individual yield when sown at the latest sowing date. Westerwolds sown at SD 3 exhibited a 1 t/ha lower grain yield than phacelia. All sowing dates had similar mean grain yields. 

Lines 95-96: “The trial site was located in Hillsborough, Co. Down, NI (Latitude 36.074600, Longitude -79.096352).” – this Hillsborough is placed in the USA, correct coordinates please.

Amended.

Lines 178-179: “The nutrient accumulation of the weeds was not added on to the results shown” - why were weeds taken into account when determining nitrogen and carbon, but not when determining nutrient assimilations? It would be logical to take weeds into account.

To determine the broad nutrient spectrum by EDXRF required a large sample for analysis compared to that to determine the C and N %. Consequently, many plots in SD1 and SD2 did not produce enough weeds to generate enough of a sample for EDXRF analysis. Consequently, this would have resulted insufficient amounts of replication and could have compromised integrity.

Lines 182-184: “Restricted maximum likelihood (REML) was used to analyse the nutrient accumulation due to the unequal number of observations." - it is also necessary to explain why REML was used to analyze LAI, barley yield with an equal number of observations.

Text inserted - Restricted maximum likelihood (REML) was used to analyse the nutrient accumulation due to the unequal number of observations as REML produces predicted means.

Line 211-213 Figure 1: Why is SED shown? It would be logical to show SEM, as in Figures 2-4.

The graphs have been modified in line with suggestions and have been changed to SEM

Lines 258-259: "Overall weed growth (measured from control plots) was 2744 kg/ha, 2271 kg/ha and 717 kg/ha at SD 1, SD 2 and SD 3, respectively” - in Fig. 2, SD2 control is clearly less than 2000 kg/ha. Why are the control strips green, despite the fact that weeds should be marked in red?

The control was highlighted green as statistically it is a cover crop so as not to confuse. I have amended the graph in line with your comment as it is technically correct.

The remaining comments are editorial.

Lines 27-28: “Keywords: cover crops; nitrogen assimilation; spring barley yield; weed management; carbon assimilation; biofertiliser; light interception carbon assimilation

Removed

Lines 46-47: “can cover crops be can be sown and grow “

Removed

Line 75: "nutrient cycling and grain yield in that pot experiment” - why “pot experiment”?

Changed to greenhouse. It is to show that this experiment is to take the findings from that previous experiment and expand upon them under field conditions. Please see text below where text in red indicates an addition of text to explain.

From a list of sixteen species of cover crops investigated by Cottney et al. [13], five species have been chosen from a range of families which showed potential to increase nutrient cycling and grain yield in that greenhouse experiment. The chosen species include forage rape (Brassica napus L.), tillage radish (Raphanus sativus L.), vetch (Vicia villosa L.), westerwolds (Lolium multiflorum L.) and phacelia (Phacelia tanacetifolia L.) to be investigated under field conditions.

Line 80: “The objective is to investigate the effect on;" - you need a colon instead of a semicolon at the end.

Changed

Line 114: ”Sowing dates (SD) were 14th August 2018 (SD 1), 7th September 2018 (SD 2), and 113 27th September 2018 (SD 3), respectively

Removed

Line 181: “Genstat Version 18 (VSN VSN International, 2017)”

Removed

Lines 187-189: "Results are deemed significant if probability due to random chance is under 5% (P<0.05) and trends are considered under 10% (P=0.10)”" - it would be correct to write “P<0.10”

Changed

Line 211-213: “Figure 1. Ceptometer readings showing the leaf area index (LAI) of the three different sowing dates for each cover crop measured at the monthly dates. Error bars represent standard error of the difference (SED). Steep declines in LAI are due to natural senescence and frost. Early sown crops were more affected by frosts than later sowing. N = 8 for each mean" – a lot of unnecessary information.

Removed in line with suggestion

Line 217: "Table 2. REML analysis of ceptometer readings" - it would be better to write “Table 2. REML analysis of leaf area index”

Amended to suggested.

Figure 2-4: "least significant difference (LSD) 0.05 = 884.7”. - mathematically incorrect “0.05=884.7". It would be better to write everywhere LSD0.05=884.7.

Changed see figures

Line 325: Figure 3 – "Error bars represent standard error of the mean (SEM) (7.89)

Removed

Lines 357-358: “(SEM) (115.3).”

Removed

Lines 391-392: “Figure 5. Nutrient assimilation of P, K and S of the cover crops in response to sowing dates (kg/ha) Slurry supplied 50 391 kg/ha P, 160 kg/ha K and 24 kg/ha S

Removed

Line 646: "4.8. Recommendations to maximize and encourage later sowing of cover crops” - it is necessary to highlight the sentence as the subtitle.

Text added to abstract:

In turn, this will help provide recommendations to maximise and encourage later sowing of cover crops.

Line 693 Table A1. TGW 2019/20 - this experience is not in the manuscript, its redundant data.

Agreed, table has been amended

Line 704: “Table A3. 4.6 2018/19 Spring barley …”

Removed

Line 736: Figure A.1 Recorded soil temperatures during both the cover crop and spring barley growth (°C - there is no closing parenthesis. Long-term average values would be interesting, as well as for precipitation on Figure A.2.

Parenthesis closed

Line 760: Figure A2 - it is necessary to specify that this is the air temperature.

Changed

Line 792: Figure A.2 is the wrong number, this is Figure A3. The name of the vertical axis "Millimeters (mm)" is incorrect, the correct one is "precipitation (mm)".

Changed

Table A.5 - in column 1, instead of SD1, SD2, SD3, it would be logical to write just 1, 2, 3, as in Tables 3 and 5.

Amended

The list of references is not designed according to the rules of the journal.

Changed in line with journal rules

Reviewer 2 Report

I find this a very timely and interesting article. It is well written and rich of contents and data. Cover crops management is one of the most important topics in agricultural sciences at the moment. Please find my comments below:

- if possible shorten the title

- the keywords should not repeat the words contained in the title

- From my point of view it is a shame that a such interesting trial has not been repeated in time. Maybe the authors should justify this choice in materials and methods.

- I found a typing error in LINE 109 "Lemkin" instead of "Lemken"

- I think the the authors, as future perspective, should think about repeating the trial within an organic farming system. Cover crops are often used in organic farms to compete with weeds and to provide nutrients to the soil. Maybe the authors should address this topic in the discussion and in the conclusion sections.

Author Response

Dear reviewer thank you for taking the time to read over the manuscript and provide feedback. This feedback has been acted upon with extensive changes highlighted both in the manuscript using track changes along with a point by point response to each comment. The point by point response to the comments is in blue writing. Furthermore, sections from the manuscript have been inserted into the response to comments with the changes made in red text whilst original text in black.

I find this a very timely and interesting article. It is well written and rich of contents and data. Cover crops management is one of the most important topics in agricultural sciences at the moment. Please find my comments below:

- if possible shorten the title

Shortened to

How cover crop sowing date impacts upon their growth, nutrient assimilation and the yield of the subsequent commercial crop

- the keywords should not repeat the words contained in the title

Modified

- From my point of view it is a shame that a such interesting trial has not been repeated in time. Maybe the authors should justify this choice in materials and methods.

The experiment was planned to be repeated but was impeded due to adverse weather. The text below has been included into 2.1 Experimental design

The experiment was planned to be repeated for two years but the second year replication was not possible due excessive rainfall.  

- I found a typing error in LINE 109 "Lemkin" instead of "Lemken"

Changed

- I think the the authors, as future perspective, should think about repeating the trial within an organic farming system. Cover crops are often used in organic farms to compete with weeds and to provide nutrients to the soil. Maybe the authors should address this topic in the discussion and in the conclusion sections.

Weed suppression is discussed in 4.7 and the following has been added to this section.

This is of particular importance in organic systems.

Reviewer 3 Report

This is an interesting and well written paper investigating the effect of five different cover crop species on the subsequent main crop spring barley.

In my review I will exclusively comment on the statistical analysis and its description.

(1) The design is a split-plot, and it is clearly stated what are the main-plot and sub-plot factors. The description of the randomization in lines 87-89, however, is rather confusing. I believe it is better to refer to two "treatment factors" rather than "treatments. Each treatment factor has several levels, and these define the actual treatments. With six cover crops (including control) combined with three sowing dates, there ought to be 3 x 5 + 1 treatments. The control requires special attention in the description of the randomization layout. Obviously, a control has no planting date, so the main question is whether or not the control was present on each whole plot. This is best stated explicitly for clarity.

(2) The reference to "blocks" in lines 87-89 is confusing. The sentence sounds as if there were two different designs during two phases of the same experiment. This would need to be very carefully explained. Also, it is unclear what exactly a "block" is here. Generally, for a description of a split-plot design, it needs to be made explicit how the whole-plot factor was randomized (e.g. completely randomized, randomized in complete blocks, etc.), and how the sub-plot factor was then randomized within whole plots. This is important because there are many variations to the split-plot design and which particular variant was chosen matters because it has implications for the appropriate statistical analysis.

(3) It appears that some traits, such as LAI, were repeatedly measured in time on the same subplots. This involves a repeated measures design and as such this should be part of the description of the design. This is also important because of the implications for analysis.

(4) The linear mixed models used for analysis should be stated explicitly so a reader can convince herself that these are correct. Different models are needed for traits measures only at one point in time and traits measured repeatedly (e.g., LAI). For the latter, the covariance structure used for modelling serial correlation needs to be spelled out. Note that such a structure is needed for both the whole-plot and sub-plot errors.

(5) It is stated in line 182 that REML was used to analyse nutrient accumulation because the data were unbalanced. I think REML could be used throughout, and the output looks as if this has actually been the case. REML is certainly needed when fitting serial correlation structures for the repeated measurements, even when the data is balanced. For traits assessed at only one time, a classical ANOVA is indeed possible for balanced data.

(6) The authors seem to be using chi-squared tests throughout (e.g. Table 2). It is much better to use F-tests. With all analyses I recommend using the Kenward-Roger method to approximate the denominator degrees of freedom, also for t-tests.

(7) The presentation of mean comparisons is not consistent across traits. For some, only the SED is presented, for others only the LSD, and for yet others no measure of presicion at all. This should be unified. If this does not seem suitable or possible, the reasons should be given in M&M.

(8) The wording explaining the meaning of letters in mean comparisons (e.g. Figure 2) is ambiguous. For a description of the problem and suggestions for less ambiguous wording see

Piepho, H.P. (2018): Letters in mean comparisons: what they do and don't mean. Agronomy Journal 110, 431-434.

(9) Table 5 lists three means for the control. But isn't this always the same treatment, as this is bare fallow? If so, one would expect to see just a single mean. This would not only be an issue in this table but in all analysis, and this comes back to the models used for analysis. The fact that there is a single control that seems to have been replicated more often than the other 15 treatments means that all comparisons with the control so far are not fully efficient. A fully efficient analysis will compute just one rather than three means for the control. This single mean will be more precise that the three separate means for the three "sowing dates". Thus, the SED for comparisons with the control could be smaller than the ones currently used. I am mindful that this kind of analysis is a bit more complex, so I do not want to impose it on the authors. But I do want to alert them to the fact that they could be doing a more efficient analysis than the one currently presented. For details, see

Piepho, H.P., Williams, E.R., Fleck, M. (2006): A note on the analysis of designed experiments with complex treatment structure. HortScience 41, 446-452.

Perhaps I am mistaken. If so, it would be useful for readers of the paper to understand why it was necessary and indeed reasonable to report three means for the control rather than one, averaged across "sowing dates."

Further comments:

Table 1: In NO2, NO3 and NH4, there are minuses and pluses placed as superscripts to the numbers 2, 3 and 4. These minuses and pluses should be superscripts on the letters O or H.

L153: 0.71 m x 0.71 m  (both sides are measured in metres)

L189: (P<0.10)

Author Response

Dear reviewer thank you for taking the time to read over the manuscript and provide feedback. This feedback has been acted upon with extensive changes highlighted both in the manuscript using track changes along with a point by point response to each comment. The point by point response to the comments is in blue writing. Furthermore, sections from the manuscript have been inserted into the response to comments with the changes made in red text whilst original text in black.

This is an interesting and well written paper investigating the effect of five different cover crop species on the subsequent main crop spring barley.

In my review I will exclusively comment on the statistical analysis and its description.

(1) The design is a split-plot, and it is clearly stated what are the main-plot and sub-plot factors. The description of the randomization in lines 87-89, however, is rather confusing. I believe it is better to refer to two "treatment factors" rather than "treatments. Each treatment factor has several levels, and these define the actual treatments. With six cover crops (including control) combined with three sowing dates, there ought to be 3 x 5 + 1 treatments. The control requires special attention in the description of the randomization layout. Obviously, a control has no planting date, so the main question is whether or not the control was present on each whole plot. This is best stated explicitly for clarity.

Due to both cultivation method and the randomisation the control was treatment and was effectively ‘sown’ at each of the dates and does have a planting date. Please see text included below and in the manuscript.

At each sowing date, the control was cultivated with the disc and left unsown as a bare fallow. This mirrors farmer practice of a stale seedbed whereby fallow land may be cultivated to both destroy and encourage more weeds to grow. 

(2) The reference to "blocks" in lines 87-89 is confusing. The sentence sounds as if there were two different designs during two phases of the same experiment. This would need to be very carefully explained. Also, it is unclear what exactly a "block" is here. Generally, for a description of a split-plot design, it needs to be made explicit how the whole-plot factor was randomized (e.g. completely randomized, randomized in complete blocks, etc.), and how the sub-plot factor was then randomized within whole plots. This is important because there are many variations to the split-plot design and which particular variant was chosen matters because it has implications for the appropriate statistical analysis.

The cover crop growth consisted of 8 blocks but was cut down to 4 during spring barley growth. The text below has been inserted into 2.1 to clarify.

The experiment was reduced in replication to 4 blocks during spring barley growth due to resource limitations.

Details have been added to explain the randomisation, see text below.

The whole plot was completely randomised with the sub-plots randomised within the whole-plots.

(3) It appears that some traits, such as LAI, were repeatedly measured in time on the same subplots. This involves a repeated measures design and as such this should be part of the description of the design. This is also important because of the implications for analysis.

The following text from section 2.42 9 (shown below) in red has been added for clarification on the model used.  Also, more detail on analysis has been inserted into the text which is also documented in response to the next comment.

2.4.2 Leaf area index (LAI)

Ceptometer readings were taken using an AccuPAR LP-80 (METER Group, Inc. Pullman, USA) which calculated the leaf area index (LAI) using a model (documented in the manual). Measurements were taken monthly, on the same sub-plots when weather conditions allowed (readings required dry and bright conditions resulting in a variation in measurement date).

(4) The linear mixed models used for analysis should be stated explicitly so a reader can convince herself that these are correct. Different models are needed for traits measures only at one point in time and traits measured repeatedly (e.g., LAI). For the latter, the covariance structure used for modelling serial correlation needs to be spelled out. Note that such a structure is needed for both the whole-plot and sub-plot errors.

The following below has been added for clarification on the model used.

The ceptometer measurements were analysed using REML as the monthly data were correlated. To account for unequally-spaced measurements, the power model of order 1 was applied to the random component (Day Number). The fixed model included was Day Number + Sowing date + Species + Day Number x Sowing date + Day Number x Species + Sowing date x Species + Day Number x Sowing date x Species, and random components were Rep + Rep x Whole plot + Rep x Whole plot x Sub plots + Plot x Day Number.

(5) It is stated in line 182 that REML was used to analyse nutrient accumulation because the data were unbalanced. I think REML could be used throughout, and the output looks as if this has actually been the case. REML is certainly needed when fitting serial correlation structures for the repeated measurements, even when the data is balanced. For traits assessed at only one time, a classical ANOVA is indeed possible for balanced data.

ANOVA has been wrongfully described due to crossover with another chapter from the thesis. You are correct it is REML used throughout. I have removed any description of ANOVA.  

(6) The authors seem to be using chi-squared tests throughout (e.g. Table 2). It is much better to use F-tests. With all analyses I recommend using the Kenward-Roger method to approximate the denominator degrees of freedom, also for t-tests.

 The data has been analysed using in-house qualified statisticians who have been advising on what they think is the most appropriate analysis method. In future studies I will consider using the Kenward-Rodger method and F-tests.

(7) The presentation of mean comparisons is not consistent across traits. For some, only the SED is presented, for others only the LSD, and for yet others no measure of precision at all. This should be unified. If this does not seem suitable or possible, the reasons should be given in M&M.

 This has been unified to ensure that each data set is presented with SEM and the LSD.

(8) The wording explaining the meaning of letters in mean comparisons (e.g. Figure 2) is ambiguous. For a description of the problem and suggestions for less ambiguous wording see

Piepho, H.P. (2018): Letters in mean comparisons: what they do and don't mean. Agronomy Journal 110, 431-434.

Thank you, this is very interest as I had not previously thought of this and I will also integrate this in future papers. Please see the amendments to the descriptions in Figure 2.

(9) Table 5 lists three means for the control. But isn't this always the same treatment, as this is bare fallow? If so, one would expect to see just a single mean. This would not only be an issue in this table but in all analysis, and this comes back to the models used for analysis. The fact that there is a single control that seems to have been replicated more often than the other 15 treatments means that all comparisons with the control so far are not fully efficient. A fully efficient analysis will compute just one rather than three means for the control. This single mean will be more precise that the three separate means for the three "sowing dates". Thus, the SED for comparisons with the control could be smaller than the ones currently used. I am mindful that this kind of analysis is a bit more complex, so I do not want to impose it on the authors. But I do want to alert them to the fact that they could be doing a more efficient analysis than the one currently presented. For details, see

Piepho, H.P., Williams, E.R., Fleck, M. (2006): A note on the analysis of designed experiments with complex treatment structure. HortScience 41, 446-452.

Perhaps I am mistaken. If so, it would be useful for readers of the paper to understand why it was necessary and indeed reasonable to report three means for the control rather than one, averaged across "sowing dates

 Due to the control being randomised in the sub-plots meant that it had to be cultivated at three different times of sowing the species. Therefore the control at sowing date 1 is different to sowing date 2 and sowing date 3 as they have been treated differently. I have put words in the text to clarify to the reader (see below).

An unplanted control of bare fallow and the 5 species of cover crops (vetch, forage rape, tillage radish, westerwolds and phacelia) were sown, Appendix (Table A.1). At each sowing date, the control was cultivated with the disc and left unsown as a bare fallow. This mirrors farmer practice of stale seedbed creation whereby fallow land may be cultivated to both destroy and encourage more weeds to grow. Hence each control at each sowing date is different to each other where each control will be presented separately in the data.

Text inserted into 2.1 Experimental design

At each sowing date, the control was cultivated with the disc and left unsown as a bare fallow. This mirrors farmer practice of stale seedbed creation whereby fallow land may be cultivated to both destroy and encourage more weeds to grow. Hence each control at each sowing date is different to each other where each control will be presented separately in the data.

Further comments: 

Table 1: In NO2, NO3 and NH4, there are minuses and pluses placed as superscripts to the numbers 2, 3 and 4. These minuses and pluses should be superscripts on the letters O or H.

Modified in line with suggestion

L153: 0.71 m x 0.71 m  (both sides are measured in metres)

 Changed

L189: (P<0.10)

Modified

VSN International, 2017. Genstat for Windows 18th, in: International, V. (Ed.). Hemel Hempstead UK.

Changed
